# Establishment of a Cell Line Stably Expressing the Growth Hormone Secretagogue Receptor to Identify Crocin as a Ghrelin Agonist

**DOI:** 10.3390/biom12121813

**Published:** 2022-12-05

**Authors:** Chia-Hao Wang, Ching-Yu Tseng, Wei-Li Hsu, Jason T. C. Tzen

**Affiliations:** 1Graduate Institute of Biotechnology, National Chung-Hsing University, Taichung 402, Taiwan; 2Graduate Institute of Microbiology and Public Health, National Chung-Hsing University, Taichung 402, Taiwan

**Keywords:** crocin, ERK phosphorylation, GHSR1a, ghrelin, lentiviral transduction, teaghrelin

## Abstract

The growth hormone secretagogue receptor-1a (GHSR1a) is the endogenous receptor for ghrelin. Activation of GHSR1a participates in many physiological processes including energy homeostasis and eating behavior. Due to its transitory half-life, the efficacy of ghrelin treatment in patients is restricted; hence the development of new adjuvant therapy is an urgent need. This study aimed to establish a cell line stably expressing GHSR1a, which could be employed to screen potential ghrelin agonists from natural compounds. First, by means of lentiviral transduction, the genome of a human HEK293T cell was modified, and a cell platform stably overexpressing GHSR1a was successfully established. In this platform, GHSR1a was expressed as a fusion protein tagged with mCherry, which allowed the monitoring of the dynamic cellular distribution of GHSR1a by fluorescent microscopy. Subsequently, the authenticity of the GHSR1a mediated signaling was further characterized by using ghrelin and teaghrelin, two molecules known to stimulate GHSR1a. The results indicated that both ghrelin and teaghrelin readily activated GHSR1a mediated signaling pathways, presumably via increasing phosphorylation levels of ERK. The specific GHSR1a signaling was further validated by using SP-analog, an antagonist of GHSR1a as well as using a cell model with the knockdown expression of GHSR1a. Molecular modeling predicted that crocin might be a potential ghrelin agonist, and this prediction was further confirmed by the established platform.

## 1. Introduction

Ghrelin, known as the hunger hormone, is composed of 28 amino acid residues esterified with a compound of octanoic acid at the third serine [1]. The growth hormone secretagogue receptor-1a (GHSR1a) has been demonstrated to be the endogenous receptor for ghrelin [2]. Recent studies have shown that the octanoyl chain of ghrelin is crucial for its access to the binding pocket of the GHSR1a receptor [3]. GHSR1a receptor is ubiquitously expressed in many regions of the brain, intestine, lung, heart, and pancreatic islets; the widespread distribution of the GHSR1a receptor implies its broad physiological effects, such as feeding behavior, reward behavior, growth hormone secretion, and memory [4,5,6]. It is known that the half-life of ghrelin in the human body is very short [7], and thus there is an urgent need for potential ghrelin agonists for the development of new adjuvant therapy. Many ghrelin agonists have been proven to be effective in the management of anorexia, sarcopenia, gastrointestinal diseases, and neurodegenerative disorders [8,9]. However, side effects were reported in patients treated with these agonists, including nausea, musculoskeletal pain, heart failure, hyperglycemia, fluid retention, and inhibition of Cytochrome P450 3A4 [10,11]. Hence, safety concern has been raised for patients treated with these ghrelin agonists.

Diverse plants with active compounds have been historically used as herbal remedies. Phytocompounds in herbal medicine are adequate candidates for the screening of novel drugs. It has been shown that 10–Gingerol isolated from *Zingiberis rhizoma* inhibited acylated ghrelin degradation enzyme and improved cisplatin-induced anorexia [12]. Anorexia caused by thermal response induced by interleukin-1β or surgery could be attenuated by sustained administration of ginsenoside Rg1, a major component of *Panax ginseng* [13,14]. Atractylodin from *Atractylodes lancea* was found to potentiate ghrelin secretion and GHSR1a receptor signaling, and thus prolonged survival in tumor-bearing rats [15]. Naringenin, a flavonoid found in a variety of herbs and fruits, has been reported to function as an agonist to the GHSR1a receptor and to inhibit weight loss [16,17,18].

Tea has played an integral role in human history and its safety is assured. Several ingredients in tea, such as epigallocatechin gallate (a catechin), caffeine, strictinin, and teaghrelin have been demonstrated to have a variety of health benefits, including the prevention of cardiovascular diseases, allergic, diabetes, cancer, and viral infections [19,20,21,22]. Teaghrelins are acylated flavonoid tetraglycoside compounds found in Chin-shin and Shy-jih-Chuen oolong tea varieties and have been shown to be the key compounds for hunger induction in drinking these tea infusions [22,23]. Accumulated studies depicted that teaghelins enhanced growth hormone secretion in primary cells, perceived to induce hunger of food intake, ameliorated dexamethasone-induced muscle atrophy in C2C12 cells, and activated the ERK1/2-pathways to antagonize MPP+ induced SH-SY5Y cells death [24,25]. Thus, teaghrelin has been proposed to be a potential oral drug with medicinal effects similar to ghrelin.

Other than teaghrelin, several candidate ghrelin agonists, including emoghrelin from Heshouwu, ginkgoghrelin from *Ginkgo biloba*, echinacoside from *Cistanche* spp., and quercetin 3-O-malonylglucoside from Mulberry (*Morus alba*) leaf, have also been identified by the same bioassay system using primary pituitary cells dissected from the anterior pituitary glands of male Sprague-Dawley rats [26,27,28,29]. To reduce animal scarification for the screening of potential ghrelin agonists from natural compounds, this study aimed to establish a cell-based system to replace the above bioassay system. A human cell line stably expressing GHSR1a was generated by means of a lentiviral system, and the authenticity of the GHSR1a mediated signaling was further examined by testing ghrelin and teaghrelin. Furthermore, crocin, a carotenoid, rich in the yellow pigment of *Gardenia jasminoides* Ellis, was shown to induce a fast-onset and prolonged antidepressant effect in prenatal stress mice presumably via GHSR-PI3K signaling [30]. On this basis, crocin was predicted as a potential ghrelin agonist by molecular modeling. To verify the modeling prediction, crocin was further examined by this cell-based platform.

## 2. Materials and Methods

### 2.1. Chemicals and Materials

Shy-jih-Chuen Oolong tea was purchased from a local tea producer in Nantou, Taiwan. All chemicals were purchased from ECHO (Miaoli, Taiwan) unless stated otherwise. Synthesized human ghrelin was purchased from Karebay Biochem (Monmouth Junction, NJ, USA). [d-Arg1, d-Phe5, d-Trp7,9, Leu11]-substance P (SP-analog), a GHSR1a antagonist, was purchased from Elabscience (Wuhan, China). Antibodies against GHSR1a and mCherry were purchased from GeneTex (Irvine, CA, USA). Antibodies against phospho-ERK1/2, ERK1/2, and tubulin were purchased from Cell Signaling (Danvers, MA, USA). Crocin was purchased from MedChemExpress (Monmouth Junction, NJ, USA).

### 2.2. Teaghrelin Isolation

Teaghrelin was extracted from Shy-jih-Chuen oolong tea according to the method described previously [23]. A tea sample of 400 g was immersed in 4 L of 50% ethanol aqueous solution for 1 h. After extraction three times, the combined filtrates were concentrated. The crude extract was purified on a Diaion HP20 (Alfa Aesar, Ward Hill, MA, USA) open column and eluted with a stepwise gradient of water and methanol (25, 50, 75, 100%) to generate four fractions. The third fraction (75%) was further purified on a Sephadex LH–20 (GE Healthcare, Chicago, IL, USA) open column eluted with water and a stepwise gradient of methanol (from 10% to 100%) to afford ten subfractions. The sixth and seventh fractions (60 and 70%) were combined and subjected to a LiChroprep^®^ RP–18 (Millipore, Merck Sigma, MA, USA) open column eluted with water and a step gradient of methanol (30, 70, and 100%) to generate three subfractions. Finally, the second fraction (70%) was collected and analyzed using high-performance liquid chromatography (Waters Corporation, Milford, MA).

### 2.3. Cell Culture and Lentiviral Transduction

The human embryonic kidney cell line 293T (HEK293T) and the lung adenocarcinoma cell line A549, purchased from the Bioresource Collection and Research Center (Hsinchu, Taiwan), were propagated in Dulbecco’s modified Eagle’s medium (DMEM) with 10% fetal bovine serum and 1% penicillin-streptomycin (Gibco, Billings, MT, USA). Cells were incubated at 37 °C with 5% CO_2_. A cell line constitutively expressing GHSR1a was generated by lentiviral transduction. In this platform, GHSR1a was expressed as a fusion protein tagged with mCherry, which allowed the monitoring of the dynamic cellular distribution of GHSR1a by fluorescent microscopy. In brief, second-generation lentiviral packaging plasmid (psPAX2), envelop plasmid (pVSVG), and the transfer plasmid with or without insertion of the gene encoding GHSR1a-mCherry was co-transfected into HEK293T at the ratio of 2:1:4 by lipofectamine 2000TM (Invitrogen, Carlsbad, CA, USA). The supernatant containing lentivirus harboring GHSR1a-mCherry gene in the transfected cells was harvested after transfection for 48 h, followed by centrifugation at 12,000× *g* to collect the virus. The concentrated virus was then used to transduce 293T cells in the presence of 8 µg/mL of polybrene (Santa Cruz, Dallas, TX, USA). After post-transduction for 24 h, HEK293T cells were selected in the presence of puromycin followed by limiting dilution and single-cell isolation monitored under fluorescent microscopy.

### 2.4. Polymerase Chain Reaction (PCR)

Total RNA was extracted from HEK293T cells with or without GHSR1a-mCherry. Total RNA (1 μg) was used to synthesize the first strand cDNA by SuperScript III (Invitrogen, Carlsbad, CA, USA), as per manufacturer’s instructions. The genotype of the modified HEK293T cells was then confirmed by PCR amplification of GHSR1a and mCherry genes from 2 μL of first strand cDNA by using specific primer sets, i.e., GHSR1a: forward 5′-TTTTGGATCCATGTGGAACGCGACGCC-3′ and reverse 5′-GGGGCTCGAGTGTATTAATACTAGATTCTGTCCAGGCC-3′; mCherry: forward 5′-TTTCTCGAGATGGTGAGCAAGGGCGAG-3′ and reverse 5′-GGGTCTAGATTACTTGTACAGCTCGTCCATGCC-3′. Briefly, the PCR conditions were conducted as follows: 5 min at 95 °C, followed by 35 cycles of 30 s at 95 °C, 30 s at 57 °C and 1 min at 72 °C. PCR products were visualized on a 1% agarose gel.

### 2.5. PrestoBlue Assay for Cell Viability

HEK293T cells harvesting GHSR1a-mCherry (2.5 × 10^4^ cells/well) or A549 cells (6 × 10^3^ cells/well) were seeded onto 96-well plates (Costar, Corning Life, Canton, MA, USA) overnight. The culture medium was replaced with serum-free DMEM media and incubated for 24 h. Cells were added with ghrelin, teaghrelin, and SP-analog at different concentrations in the presence of 10% PrestoBlue (Invitrogen, Carlsbad, CA, USA) and incubated for 4 h. Then the absorb wavelength at 570 nm was measured by Tecan infinite 200 PRO spectrophotometer (Tecan Group Ltd., Männedorf, Switzerland). Cell viability was calculated as the percentage of control (cells in 1× Hanks balanced salt solution) (Gibco).

### 2.6. Immunoblotting Analysis

GHSR1a-mCherry-HEK293T (1.25 × 10^6^ cells/well) or A549 (3 × 10^5^ cells/well) cells were seeded onto 12-well plates (Costar, Corning Life, Canton, MA, USA). Cells were incubated with ghrelin and teaghrelin at different concentrations of up to 100 µM for 15 to 60 min. Cells were lysed in lysis buffer (150 mM NaCl, 1.0% Triton X-100, 50 mM Tris-Cl, adjust pH to 7.4) supplemented with protease inhibitor cocktail and phosphatase inhibitor (Roche, Basel, Switzerland). Cell lysates were clarified by centrifugation for 15 min at 15,000× *g* at 4 °C. Proteins were separated by SDS-PAGE (5% stacking gel, 12% separating gel) and electroblotted onto PVDF membrane (GE Healthcare, Chicago, IL, USA). The membranes were incubated for 10 min at room temperature in PRO blocking buffer (Visual Protein, Energenesis Biomedical, Taipei, Taiwan), followed by overnight incubation with antibodies against pERK1/2, ERK1/2, mCherry, GHSR1a, or tubulin at 4 °C. On the following day, the membranes were incubated with the HRP-labeled peroxidase AffiniPure goat anti-rabbit IgG (1: 5000; Jackson ImmunoResearch Laboratories, West Grove, PA, USA) for 1 h at room temperature. Blots were then visualized with the Immobilon western chemiluminescent HRP substrate (Millipore, Merck Sigma, MA, USA).

### 2.7. Immunofluorescence Staining

Cells plated onto µ-Slide 8 Well (Ibidi, Martinsried, Germany) were fixed in 1% paraformaldehyde, and then incubated with the anti-GHSR1a antibody (GeneTex, Irvine, CA, USA), followed by incubation with anti-rabbit IgG Alexa Fluor™ 488 (Invitrogen, Carlsbad, CA, USA). The nuclei were counterstained with Hoechst 33342 (Invitrogen, Carlsbad, CA, USA), and the cells were visualized in fluorescence microscopy (FV3000; Olympus Corporation, Tokyo, Japan).

### 2.8. Silence of GHSR1a by RNA Interference

Small interfering RNA (siRNA) technology was used to silence the GHSR1a expression in A549 cells. Cells were seeded in 12-well culture plates at a density of 3 × 10^5^ cells/well for 24 h. GHSR1a-siRNA was transfected into cells by TransIT-X2^®^ (Mirus Bio, Madison, WI, USA) according to the manufacturer’s instructions. The transfected cells were cultured at 37 °C and 5% CO_2_ for another 24 h. The interference efficiency of siRNA was detected by immuno-blotting followed by the measurement of the GHSR1a signal using Image J. GHSR1a-siRNA sequences were as follows, Forward primer: 5′-CGTCACGTTGGACCTGGATTTCAAGAGAATCCAGGTCCAACGTGACGTTTTTT-3′, Reverse primer: 5′-AATTAAAAAACGTCACGTTGGACCTGGATTCTCTTGAAATCCAGGTCCAACGTGACGGGCC-3′.

### 2.9. Homology Modeling and Docking

Homology modeling and docking to GHSR1a were performed according to the procedure described in a previous study [23]. Homology modeling and docking were performed on growth hormone secretagogue receptor 1a (GHSR1a, accession number AAI13548). The crystal structures of β1 and β2 adrenergic receptor binding ligands were used to construct the GHSR structure through homology modeling. The 3D structures of GHRP-6 (a synthetic GHSR agonist), teaghrelin, and crocin were constructed in the Chem3D program (http://www.cambridgesoft.com/ accessed on 23 November 2022). Interaction between GHSR1a and GHRP-6, teaghrelin, or crocin was simulated within the spherical space of a 14 Å radius from the center of the binding pocket. Molecular docking was performed using the Discover Studio 2.1 package and further minimized by a smart minimize algorithm with a CHARMm force field in the Discover Studio 2.1 package [31]. To compare the relative binding affinities of teaghrelin and crocin in GHSR1a, the binding site of GHSR1a was used for the docking to calculate the binding energy by GEMDOCK (The Institute of Bioinformatics, National Chiao Tung University, Hsinchu, Taiwan).

### 2.10. Statistical Analysis

The data were presented as mean values ± standard error of the mean (SEM). The differences were analyzed by one-way analysis of variance (ANOVA) followed by Tukey’s test. Statistical calculations were performed by GraphPad Prism 7 (GraphPad Prism Software, San Diego, CA, USA). Differences of *p* < 0.05 were deemed significant. The number of independently performed experiments was described in the figure legends.

## 3. Results

### 3.1. Generation of Human HEK293T Cells Expressing GHSR1a

To facilitate the observation of GHSR1a expressed in human HEK293T cells, GHSR1a was genetically engineered to tag with mCherry, a fluorescence protein at its C terminus. Cells expressing the GHSR1a-mCherry protein were selected under a fluorescent microscope and further isolated for homogeneity. The presence of the GHSR1a-mCherry gene in genetically modified cells, namely GHSR1a-mCherry HEK293T, was validated by PCR. As expected, PCR products with expected sizes encoding for GHSR1a, mCherry, and GHSR1a-mCherry, were successfully amplified from corresponding cells (Figure 1A). In fluorescence microscopy, mCherry was found co-localized with the signal yielded by GHSR1a specific antibody (Figure 1B) as measured with Pearson’s coefficient using Image J (JACoP model, Overlap Coefficient: r = 0.658). The data indicated that GHSR1a was expressed as a fusion protein with mCherry in GHSR1a-mCherry HEK293T cells. Accordingly, the GHSR1a-mCherry protein of expected molecular mass was detected by western blot analysis using the antibody against mCherry (Figure 1C).

### 3.2. Effects of Ghrelin, Teaghrelin, and SP-Analog on Cell Viability of GHSR1a-mCherry HEK293T and A549 Cells

Effects of two GHSR1a agonists, ghrelin and teaghrelin as well as those of an antagonist, SP-analog on cell viability of GHSR1a-mCherry HEK293T and A549 cells were examined (Figure 2). Cells were treated with ghrelin, teaghrelin, and SP-analog of different concentrations for 4 h, and their viability was depicted by PrepstoBlue assay. The results indicated that no significant impact on cell viability was observed for ghrelin (up to 100 nM), teaghrelin (up to 100 μM), and SP-analog (up to 5 μM).

### 3.3. Activation of Intracellular ERK1/2 Signaling in GHSR1a-mCherry HEK293T Cells

As engagement of ghrelin with the GHSR1a receptor was shown to trigger ERK signaling pathways [32,33], the authenticity of GHS-R1a was validated by evaluation of the ERK1/2 status in GHSR1a-mCherry HEK293T cells. Upon ghrelin (100 nM) treatment, ERK1/2 phosphorylation level was remarkably increased in GHSR1a-mCherry HEK293T cells as compared with the untreated cells or wild-type HEK293T cells (Figure 3A). Subsequently, the activation status of ERK1/2 in GHSR1a-mCherry HEK293T cells was comparatively analyzed under the stimulation of ghrelin and teaghrelin with or without the company of SP-analog. Compared with the control, ghrelin at concentrations of 10 nM and 100 nM remarkably increased the level of ERK1/2 phosphorylation (p-ERK1/2) (Figure 3B). Similarly, teaghrelin, ranging from 5 to 100 μM, enhanced ERK1/2 phosphorylation in a dose-dependent manner (Figure 3C). The enhancement of ERK1/2 phosphorylation by ghrelin or teaghrelin was greatly suppressed in GHSR1a-mCherry HEK293T cells when SP analog was accompanied (Figure 3D,E).

### 3.4. Activation of Intracellular ERK1/2 Signaling in A549 Cells

Human lung carcinoma A549 cell, shown to express GHSR1a, has been extensively exploited to study the GHSR1a mediated effect [34]. The strategy used for validation of the authenticity of GHSR1a-mCherry HEK293T cells was also tested on A549 cells. The signaling of endogenous GHSR1a receptor under stimulation of ghrelin or teaghrelin in A549 cells was explored. Consistently, the level of phosphorylated ERK1/2 in A549 cells was increased upon treatment of ghrelin (Figure 4A) and teaghrelin (Figure 4B) in a dose-dependent manner. It was worth noting that the highest level of ERK1/2 activation was detected at 15 min of ghrelin treatment and it was declined afterward; the activation of ERK1/2 ceased at 60 min treatment. In contrast, teaghrelin treatment continuously enhanced ERK1/2 activation along the time, and the highest level was observed at 60 min treatment.

To investigate whether the ERK1/2 phosphorylation induced by ghrelin and teaghrelin was mediated by the GHSR1a receptor in A549 cells, an attempt was made to block the GHSR1a receptor function by using the antagonist of the GHSR1a receptor, SP-analog. Similar to the observation in GHSR1a-mCherry HEK293T cells, pretreatment of SP-analog indeed suppressed phosphorylation of ERK1/2 elicited by ghrelin (Figure 4C) or teaghrelin (Figure 4D). To further confirm that ghrelin- and teaghrelin-induced ERK1/2 phosphorylation was specifically mediated by GHSR1a receptor, GHSR1a expression was knocked down by the RNA interference technique. As compared with the GHSR1a level in the BB group, siRNA treatment reduced the protein level of GHSR1a by approximately 70% (Figure 4C). Moreover, the knockdown of GHSR1a expression significantly attenuated the ERK1/2 phosphorylation in A549 cells induced by ghrelin (Figure 4C) or teaghrelin (Figure 4D). Noticeably, the level of ERK1/2 in cells with knockdown of GHSR1a expression was similar to that pretreated with SP-analog.

### 3.5. Modeling of Crocin Docking to the GHSR1a Receptor

In addition to GHRP-6 and teaghrelin, crocin was subjected to molecular modeling and docking to the binding pocket of the GHSR1a receptor. The results showed that crocin could get into and interact with the binding pocket of GHSR1a, although its interactions with the receptor were relatively weak in comparison with GHRP-6 (Figure 5). In addition to π-π interaction, eight H-bonds were formed between teaghrelin and the GHSR1a receptor. Only seven H-bonds were identified for the interaction between crocin and GHSR1a. To quantitatively estimate the relative strength of interaction between teaghrelin or crocin and the GHSR1a receptor, chemical energy was calculated by GEMDOCK, and the results indicated that the strength of ligand-receptor interaction between crocin and GHSR1a was slightly weaker than that between teaghrelin and GHSR1a (Table 1). Taken together, crocin was predicted to be a ghrelin agonist.

### 3.6. Detection of Crocin as a Ghrelin Agonist

To verify the prediction of molecular modeling, crocin was examined for its potential as a ghrelin agonist by the cell line stably expressing GHSR1a. No significant impact on cell viability was observed for crocin at a concentration of up to 100 μM (Figure 6A). Upon ghrelin (10 nM) treatment, ERK1/2 phosphorylation level was remarkably increased in the cells, and a significant increase of ERK1/2 phosphorylation level was also observed when the cells were treated with 5, 10, 50, and 100 μM of crocin (Figure 6B,C). The results were in agreement with the modeling prediction indicating that crocin might be used as a weak ghrelin agonist.

## 4. Discussion

As the GHSR1a receptor is mainly distributed in the brain, pituitary cells isolated from rats have been used as an in vitro system to investigate GHSR1a receptor-mediated growth hormone secretion [34]. Nevertheless, this method requires constant re-acquisition of fresh cells from animals. To reduce unnecessary animal sacrifices, we aimed to generate a stable cell line expressing the GHSR1a receptor to replace primary pituitary cells dissected from the anterior pituitary glands of rats for the screening of potential ghrelin agonists. Owing to its high transfection efficiency and satisfactory protein yields, HEK 293 cell has been commonly used to express heterogeneous proteins, including oxytocin receptor [35], glucagon-like peptide-1 receptor [36], and dopamine receptor type 2 [37]. Many reports have explored the physiological activities of ghrelin using the HEK293T cell line transiently expressing the GHSR1a receptor [33,38,39,40]. However, the transient expression could be cumbersome owing to the variable transfection efficiency from batch to batch and also the possible deleterious effect to cells which might affect cell signaling. Moreover, transiently transfected cells tend to express the receptor in a limited period of time. Therefore, in this study, we generated a cell line, GHSR1a-mCherry HEK293T, which stably expressed the GHSR1a receptor at high level, and confirmed that this cell line could be sensitized upon the stimulation of ghrelin, the endogenous ligand of the GHSR1a.

It has been shown that activation of the GHSR1a receptor by ghrelin led to the stimulation of ERK signaling [41,42]. Thus, the authenticity of the GHSR1a-mCherry HEK293T cell line was also validated by monitoring the phosphorylation level of ERK1/2. The results showed that ERK1/2 phosphorylation was triggered under ghrelin treatment in a dose-dependent manner, and the enhancement of ERK1/2 phosphorylation was abolished by adding the antagonist, SP-analogue (Figure 4). Furthermore, teaghrelin was also demonstrated to activate ERK1/2 singling by binding to the GHSR1a receptor. In consistence with the previous studies [25], the effect of teaghrelin on GHSR1a-mCherry HEK293T cells was diminished by the treatment of SP-analogue. On the basis of the induction of the growth hormone secretion from rat pituitary cells, teaghrelin was found 1000 times weaker than GHRP-6 [22,23]. According to this study, teaghrelin was estimated to be 10,000 times weaker than ghrelin regarding their relative activities to induce ERK1/2 activation.

Ghrelin has been reported to efficiently induce ERK1/2 phosphorylation in A549 cells [34]. In this study, teaghrelin was observed to slowly induce ERK1/2 phosphorylation in A549 cells in comparison with ghrelin (Figure 4A,B). Although the specificity of the ERK1/2 mediated reaction in the GHSR1a-mCherry HEK293T cell has been validated by using SP-analog (Figure 4C,D), it may not be a bona fide inverse agonist. SP-analog has been identified as a potent antagonist of the bombesin and mammalian gastrin-releasing peptide, mitogens for Swiss 3T3 cells [43]. Moreover, SP-analog is an endogenous agonist for the neurokinin-1 receptor, which is a transmembrane protein classified into the family of G-protein-coupled receptors [44]. It could be argued that the effect of SP-analog observed in the present study might not be simply mediated by the GHSR1a receptor. Therefore, the knockdown of GHSR1a expression was also used to confirm that the GHSR1a receptor-mediated signaling was indeed activated by ghrelin in A549 cells. As shown in Figure 4, the overall GHSR1a expression was decreased under siRNA treatment and that also led to the reduction of ERK1/2 phosphorylation level induced by ghrelin. The results strengthened the authenticity of GHSR1a mediated ERK1/2 activation in our system.

It has been well documented that ghrelin secretion and expression of GHSR1a was relatively high in various types of tumors including pituitary adenomas, other neuroendocrine cancers, breast carcinomas, and prostate cancer cell lines [45]. Hence, alternatively, cancerous cells, such as human lung carcinoma A549 cells [46], might serve as an in vitro model to detect the downstream ERK signaling upon activation of the GHSR1a receptor [34]. As illustrated in Figure 4A, the peak of ERK activation occurred at 15 min post-induction, indicating an efficient signal transduction triggered by ghrelin. However, it should be noticed that the levels of GHSR1a receptor expression in cancer cells are variable and the cellular signaling in a tumor may not be the same as in normal cells, possibly leading to inconsistent effects between tumor and normal cells. Therefore, the HEK293T cell line, instead of the A549 cell line, expressing the GHSR1a receptor will be employed to screen potential ghrelin agonists from natural compounds in our follow-up research.

*Gardenia jasminoides* Ellis has been used as an herbal medicine for thousands of years, and daily consumption of its fruit was believed to calm the mind, improve the quality of sleep, and enhance digestive systems [47]. Crocin, a water-soluble carotenoid in this herb, has been identified as an active ingredient responsible for several pharmacological effects on neurodegeneration, cardiovascular disease, cerebrovascular disease, depression, and liver disease [48]. Recently, crocin was shown to induce a fast-onset and prolonged antidepressant effect in prenatal stress mice, and the rapid and prolonged antidepressant-like effect of crocin was presumably activated via GHSR-PI3K signaling [30]. Accordingly, crocin was assumed to be a potential ghrelin agonist. In this study, molecular modeling suggested that crocin might be a ghrelin agonist, and this prediction was further confirmed by the established cell-based platform.

## 5. Conclusions

We have successfully established a reliable cell platform that stably expresses the GHSR-1a receptor. As validated by induction of the authentic ligand (ghrelin) and a natural compound (teaghrelin), the corresponding cellular signaling was elicited in this cell system, and the activation of cellular signaling was counteracted by treatment of an antagonist, SP-analog as well as by siRNA against GHSR1a. This report cell model seems to be a user-friendly system for high throughput screening the ghrelin-mimicking compounds for therapeutic purposes. As an example, crocin, a candidate ghrelin agonist, was confirmed by this cell platform.

## Figures and Tables

**Figure 1 biomolecules-12-01813-f001:**
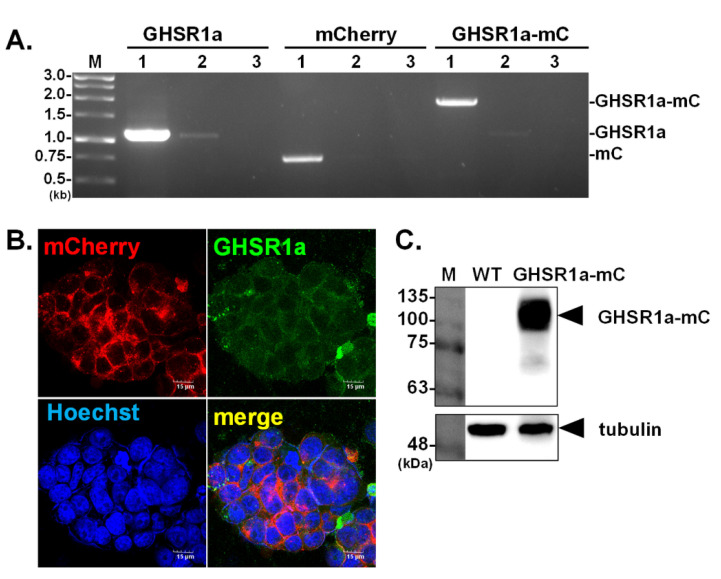
Establishment of HEK293T cells stably expressing the GHSR1a receptor fused with mCherry. The presence of GHSR1a, mCherry (mC), and GHSR1a-mCherry gene segments in GHSR1a-mCherry HEK293T (lane 1) and wild-type (WT) HEK293T (lane 2) cells was verified by PCR (**A**); lane 3 is negative control. The cellular distribution of GHSR1a-mCherry was examined in confocal microscopy. Co-localization of the GHSR1a-mCherry (red) with the GHSR1a receptor (green) was observed. Hoechst staining indicated the location of the nucleus (**B**). The whole cell lysate was harvested for detection of GHSR1a-mCherry expression by western blot analysis using the antibody against mCherry (**C**).

**Figure 2 biomolecules-12-01813-f002:**
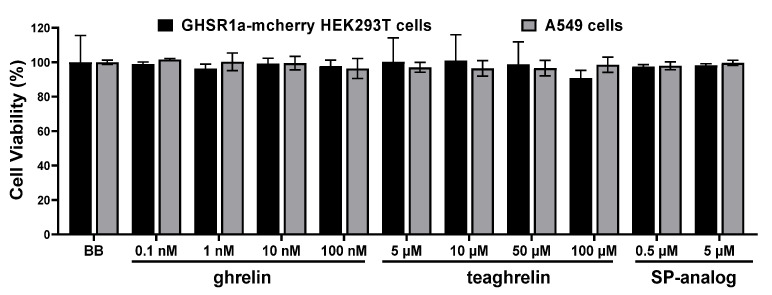
Cell viability of GHSR1a-mCherry HEK293T and A549 cells in the presence of ghrelin, teaghrelin, and SP-analog. Cells were treated with ghrelin, teaghrelin, and SP-analog at indicated concentrations followed by the PrepstoBlue assay. Data were expressed as the percentage of viability with respect to the control. Graphs represented the mean ± SD of triplicate samples.

**Figure 3 biomolecules-12-01813-f003:**
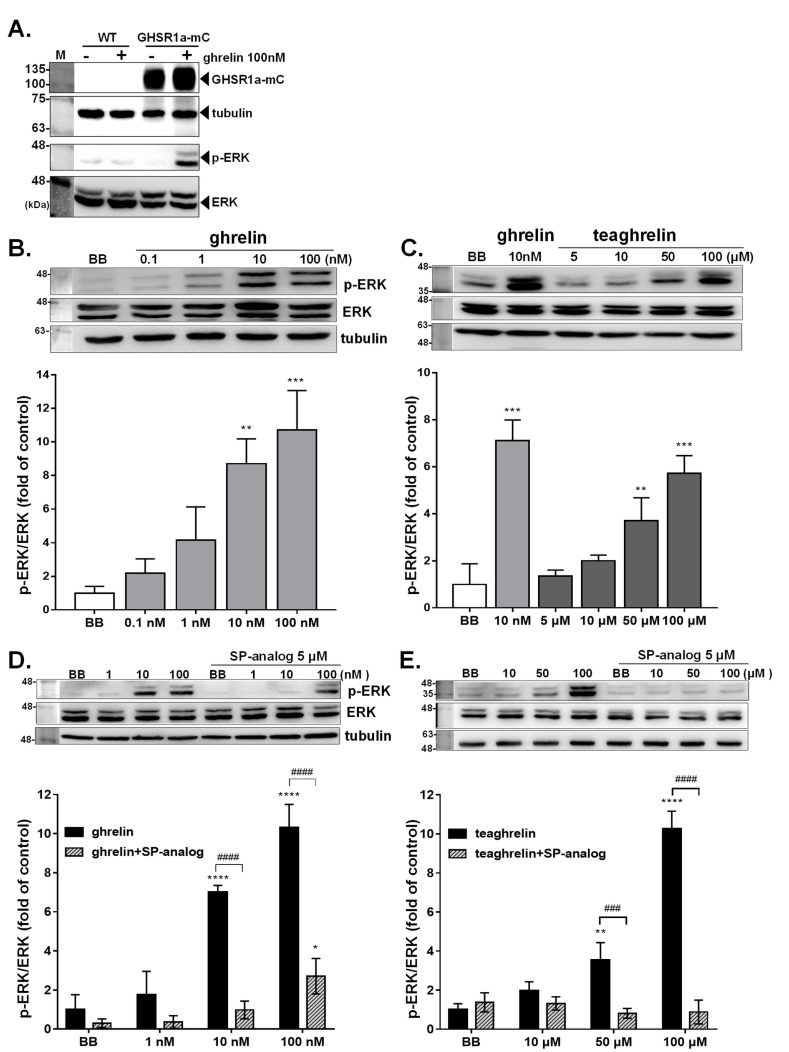
Effects of ghrelin and teaghrelin on ERK1/2 phosphorylation in GHSR1a-mCherry HEK293T cells. The wild-type HEK293T or GHSR1a-mCherry HEK293T cells were treated with or without ghrelin (100 nM) for 1 h, and the ERK1/2 activation level was detected by western blot analysis (**A**). The effects of ghrelin and teaghrelin on ERK1/2 activation status in cells were detected without (**B**,**C**) or with the company of the antagonist SP-analog (**D**,**E**); BB represented the buffer control without ghrelin, teaghrelin, and SP-analog. The activation status of ERK1/2 was detected by western blot analysis with α-tubulin as a loading control. Data from three independent repeats were analyzed, and the relative activation level was plotted with means ± SD. * *p* < 0.05, ** *p* < 0.01, *** *p* < 0.001, **** *p* < 0.0001, compared with the control group. ^###^ *p* < 0.001, ^####^ *p* < 0.0001, compared with the treatment with ghrelin or teaghrelin.

**Figure 4 biomolecules-12-01813-f004:**
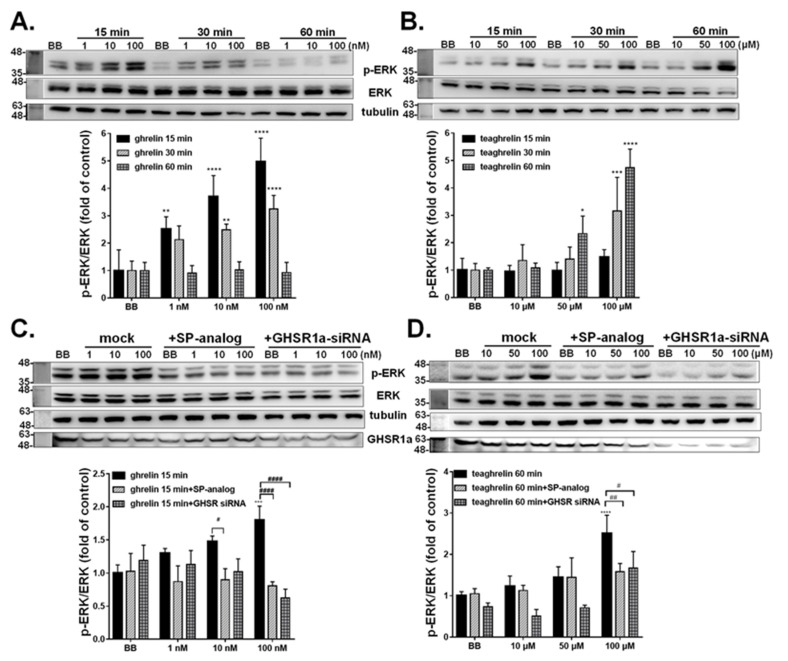
Effects of ghrelin and teaghrelin on ERK phosphorylation in A549 cells. Ghrelin (**A**) and teaghrelin (**B**), at indicated concentrations, were administrated in A549 cells for 15, 30, and 60 min; BB represented the buffer control. Moreover, A549 cells were pretreated with SP-analog (30 min) or transiently transfected with GHSR1 siRNA followed by treatment of ghrelin (**C**) and teaghrelin (**D**) for 15 min or 60 min, respectively. The activation status of ERK1/2 was detected by western blot analysis with α-tubulin as a loading control. Data from three independent repeats were analyzed and the relative activation level was plotted with means ± SD. * *p* < 0.05, ** *p* < 0.01, *** *p* < 0.001, **** *p* < 0.0001, compared with the control group. ^#^ *p* < 0.05, ^##^ *p* < 0.01, ^####^ *p* < 0.0001, compared with the treatment with ghrelin or teaghrelin.

**Figure 5 biomolecules-12-01813-f005:**
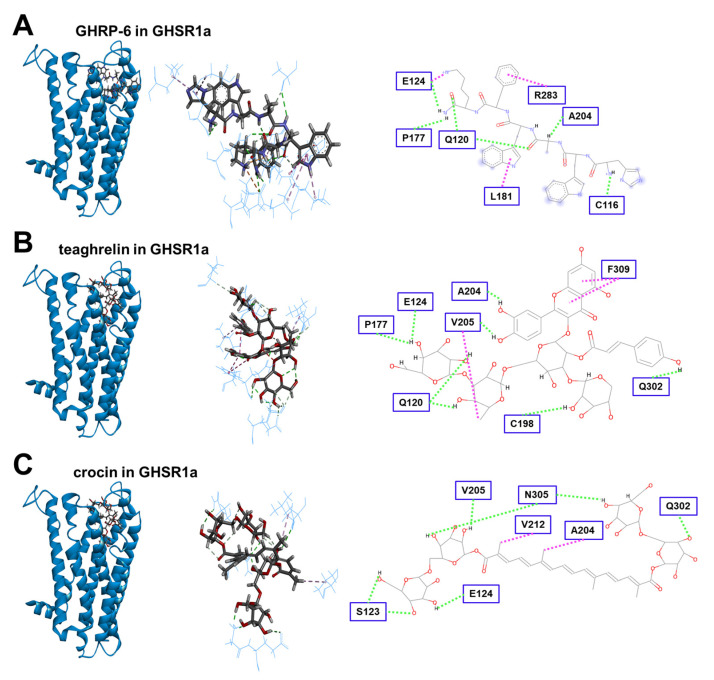
Modeling of GHRP-6 (**A**), teaghrelin (**B**), and crocin (**C**) docking into GHSR1a. Left panel: ligands (ball-stick structure) in the binding site of GHSR1a (ribbon structure). Middle panel: the amino acids (stick structure) close to ligands (ball-stick structure) in these complex structures. Right panel: detailed intermolecular interaction between ligands and GHSR1a. The amino acids of GHSR1a involved in the formation of interaction with ligands are shown in squares. H-bonding (green lines) and π-π interaction (pink lines) are indicated.

**Figure 6 biomolecules-12-01813-f006:**
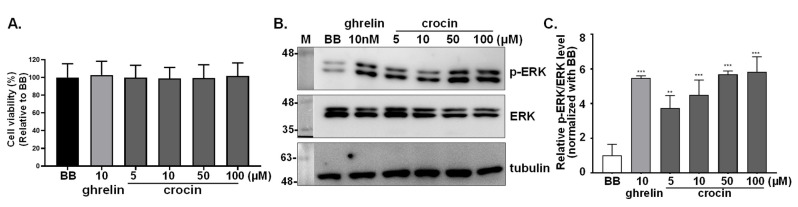
(**A**) Cell viability of GHSR1a-mCherry HEK293T cells in the presence of crocin. Cells were treated with crocin of 5, 10, 50, and 100 μM followed by the PrepstoBlue assay; BB represented the buffer control. Data were expressed as the percentage of viability with respect to the control. Graphs represented the mean ± SD of triplicate samples. (**B**) Effects of crocin on ERK1/2 phosphorylation in GHSR1a-mCherry HEK293T cells. The effects of crocin on ERK1/2 activation status in cells were detected by western blot analysis with α-tubulin as a loading control. (**C**) Statistical analysis of crocin effects on ERK1/2 activation. Data from three independent repeats were analyzed, and the relative activation level was plotted with means ± SD. ** *p* < 0.01, *** *p* < 0.001, compared with the control group.

**Table 1 biomolecules-12-01813-t001:** Chemical energy calculated by GEMDOCK for the interaction between teaghrelin or crocin and the binding pocket of the GHSR1a receptor.

Ligand	Total Energy (kJ mol^−1^)	Van der Waals’ Force (kJ mol^−1^)	H Bond (kJ mol^−1^)
teaghrelin	−157.97	−17.87	16.63
crocin	−146.43	−15.65	15.97

## Data Availability

Not applicable.

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
