# Peer review of "Establishment of a Cell Line Stably Expressing the Growth Hormone Secretagogue Receptor to Identify Crocin as a Ghrelin Agonist"

_biomolecules, 2022, doi:10.3390/biom12121813_

Round 1
Reviewer 1 Report
Overall the significance of the article is reduced substantially by lack of new drug screening with the model system developed.
In figure 1, the co-localization is not clear visually, authors should consider using a metric to find percent co-localization.
HEK-293 cells are immortalized cells, how would that effect its comparison to normal cells.
What is the purpose of a tag in finding an agonist.
All westernblots require size markers and quantification plots.
The article failed to compare the levels of receptor overexpressed to normal types of tissue. 293 cells appear to express higher levels of receptor compared to levels observed in normal tissue. This could effect drug screening.
Reviewer 2 Report
Although this manuscript established an interesting method for screening ghrelin agonists, I didn't find technical novelty when compared with other commercially available methods such as CHO-K1/GHSR stable cell line. In addition, they did not discuss the advantages of their cell line compared to other cell lines. Considering the Journal Biomolecules Impact Score, although the data on p-ERK are clear, I think that identifying one target molecules is not sufficient for publication.
In addition, where is the data showing 70% GHSR1a knockdown?
Round 2
Reviewer 1 Report
I have no further comments, minor grammar check is required.
Author Response
Thanks to the reviewers for their comments.
Reviewer 2 Report
The questions raised were fully explained.
Author Response

(The authors gave the same response as above.)
